# Association between serum uric acid and spirometric pulmonary function in Korean adults: The 2016 Korea National Health and Nutrition Examination Survey

**Jae Won Hong, Jung Hyun Noh, Dong-Jun Kim** * 

Department of Internal Medicine, Ilsan-Paik Hospital, College of Medicine, Inje University, Koyang, Gyeonggi-do, Republic of Korea

* djkim@paik.ac.kr

## Abstract

### Background

A limited number of epidemiological studies have investigated the association between serum uric acid and pulmonary function in the general population. However, the results have been inconclusive.

### Objectives

This study was performed to investigate the association between serum uric acid and spirometric pulmonary function in general population.

### Methods

Among the 8,150 participants who participated in the 2016 Korea National Health and Nutrition Examination Survey, 2,901 participants were analyzed in this study. Subjects were divided into four groups according to forced vital capacity (FVC)% predicted or forced expiratory volume in 1 second (FEV1) % predicted quartiles. Participants in the lowest quartile of FVC % predicted and FEV1% predicted were compared to those in the remaining quartiles according to age, education level, household income, smoking status, alcohol consumption, aerobic exercise, obesity, hypertension, diabetes, renal impairment, serum uric acid, and hyperuricemia. Multivariable logistic regression analyses were used to calculate the odds ratio (OR) of hyperuricemia for participants in the lowest quartile of FVC% and FEV1 predicted, with above covariates.

### Results

In women, hyperuricemia was associated with lowest quartile of FVC% predicted (OR 1.71, 95% CI 1.06–2.75, $p = 0.027$) and FEV1 predicted (OR 1.70, 95% CI 1.06–2.74, $p = 0.028$) respectively, serving as above confounding variables. In men, hyperuricemia (OR 1.54, 95% CI 1.07–2.22, $p = 0.021$) was associated with the lowest quartile of FEV1% predicted, not FVC% predicted.

**Data Availability Statement:** Readers can access the dataset by registering an account with the Korean CDC website (https://knhanes.cdc.go.kr/knhanes/index.do). There is a blue bar on the top of

the website. Click the third menu, written in Korean, "원시자료" (The content about the raw data) on the blue bar. There is a second submenu below the blue bar, written in Korean "원시자료 다운로드 (Download the raw data)". Once readers click this, an e-mail address for log-in is required. Once logged in, readers can download the raw data from 1998-2016 Korea National Health and Nutrition Examination Survey database using SAS or SPSS. The authors do not have any special access privileges to the data. For other data related inquiries, please contact the corresponding author.

**Funding:** The author(s) received no specific funding for this work.

**Competing interests:** The authors have declared that no competing interests exist.

According to median age, in women, age $\geq$ 56 years old with hyperuricemia was associated with lowest quartile of FVC% predicted (OR 1.85, 95% CI 1.04–3.28, $p = 0.037$) and FEV1% predicted (OR 1.99, 95% CI 1.11–3.75, $p = 0.021$), respectively. In men, age $\geq$ 56 years old with hyperuricemia was associated with lowest quartile of FEV1% predicted (OR 1.75, 95% CI 1.05–2.94, $p = 0.033$), not FCV% predicted.

## Conclusions

Hyperuricemia was associated with lowest quartile of FEV1% or FVC% predicted in Korean general population. This correlation between hyperuricemia and low pulmonary function was more pronounced in women and older age.

## Introduction

Uric acid is primarily synthesized in the liver, intestines, muscles, kidneys and vascular endothelium, as an end product of the action of xanthine oxidase on exogenous purines [1]. Hyperuricemia is well known to be associated with increased risk of incident and recurrent gout [2]. However, recent research in uric acid has focused primarily on its potential as a mediator of human diseases other than gout.

Most epidemiologic studies have suggested that elevated serum uric acid levels are associated with cardiovascular diseases, including coronary heart disease, stroke, congestive heart failure, and hypertension [3–6]. Furthermore, traditional cardiovascular risk factors, including metabolic syndrome, insulin resistance, obesity, non-alcoholic fatty liver disease, and chronic kidney disease, are also related to elevated serum uric acid levels [7–11]. Several experimental and clinical studies support a role of uric acid as a causal factor in these conditions, via its contributions to systemic inflammation, endothelial dysfunction, and oxidative stress [1, 8].

In this context, increased levels of uric acid have also been observed in a variety of respiratory disorders, including obstructive sleep apnea, pulmonary hypertension and chronic obstructive pulmonary disease (COPD) [12–14]. A limited number of epidemiological studies have investigated the association between serum uric acid levels and pulmonary function in the general population [15, 16]. However, the results have been inconclusive.

In this study, we investigated the association between serum uric acid and pulmonary function in Korean population using data from the 2016 Korea National Health and Nutrition Examination Survey (KNHANES).

## Methods

### Study population and data collection

This study was based on data from the 2016 KNHANES, a cross-sectional and nationally representative survey conducted by the Korean Center for Disease Control for Health Statistics. The KNHANES has been conducted periodically since 1998 to assess the health and nutritional status of the noninstitutionalized civilian population of Korea. Participants were selected using a systemic proportional allocation sampling method with multistage stratification. A standardized interview was conducted in the homes of all participants to collect information on demographic variables, family history, medical history, medications used, and a variety of other health-related variables. The health interview was based on an established questionnaire to determine the demographic and socioeconomic characteristics of the subjects including

age, education level, occupation, income, marital status, smoking status, alcohol consumption, aerobic exercise, previous and current diseases, and family disease history.

As part of the interview, subjects were asked whether they exercised with an intensity that caused a slight increase in breathing rate and sweating; those who exercised regularly at moderate intensity were asked about the frequency with which they exercised per week, and the length of time per exercise session. Participants who performed aerobic exercise were defined as those who spent at least 150 min performing moderate physical activity, or 75 min performing high-intensity physical activity, per week. Alcohol consumption was assessed based on responses to questions regarding drinking behavior, including average alcohol consumption and drinking frequency, during the month prior to the interview. Diabetes was defined as a fasting plasma glucose (FPG) concentration $\geq$ 126 mg/dL (7.0 mmol/L), current use of anti-diabetes medication, or a previous diagnosis of diabetes by a physician. Obesity was defined as a body mass index (BMI) $\geq$25 kg/m$^2$ according to the Asia-Pacific obesity classification [17]. Height and weight were obtained using standardized techniques and equipment. Height was measured to the nearest 0.1 cm using a portable stadiometer (Seriter, Bismarck, ND, USA). Weight was measured to the nearest 0.1 kg using a Giant-150N calibrated balance-beam scale (Hana, Seoul, Korea). BMI was calculated by dividing the weight by the height squared (kg/m$^2$). Systolic and diastolic blood pressure were measured by standard methods using a standard mercury sphygmomanometer (Baumanometer, WA Baum Co. Inc., Copiague, NY, USA), while the patient was seated. Three measurements were obtained for all subjects at 5-min intervals, and the average of the second and third measurements was used in the analysis. Hypertension was defined as systolic blood pressure $\geq$ 140 mmHg, or diastolic blood pressure $\geq$ 90 mmHg, or use of antihypertensive medications irrespective of blood pressure.

Of the 8,150 participants in the 2016 KNHANES, the number of individuals aged $\geq$ 19 years were 6,382. Among these subjects, 3,341 were assessed for both serum uric acid levels and pulmonary function. Pulmonary function tests were performed only in individuals aged $\geq$ 40 years.

A total of 282 subjects were excluded from the analysis due to pre-existing diseases, including liver cirrhosis ($n$ = 13), renal failure ($n$ = 9), lung cancer ($n$ = 9), asthma ($n$ = 94), heart failure ($n$ = 117), and cerebrovascular accident ($n$ = 60). A total of 158 examinees who had baseline FEV1 or FVC quality attribute of "D (questionable results, use with caution)" or "F"(results not valid)" were also exclude [18, 19]. Finally, 2,901 participants were analyzed in this study (Fig 1).

## Pulmonary function test

Well-trained technicians measured forced vital capacity (FVC) and forced expiratory volume in 1 second (FEV1) using a spirometer (Vyntus Spiro; Care Fusion, San Diego, CA, USA) and SentrySuite (Care Fusion) according to the American Thoracic Society criteria [20]. All spirometry values were prebronchodilator results.

The FVC% predicted and FEV1% predicted were calculated by dividing the FVC and FEV1 by the predicted FVC and FEV1, respectively. The predicted FEV1 and FVC for each subject were calculated using published equations [21]. Subjects were divided into four groups according to FVC% predicted or FEV1% predicted quartiles, as follows:

FVC% predicted:

Men: Q1 <82.4%; Q2 82.–89.9%; Q3 90.0–97.2%; Q4 $\geq$ 97.3%

Women: Q1, <84.1%; Q2 84.2–91.8%; Q3 91.9–100.0%; Q4 $\geq$ 100.1%

FEV1% predicted:

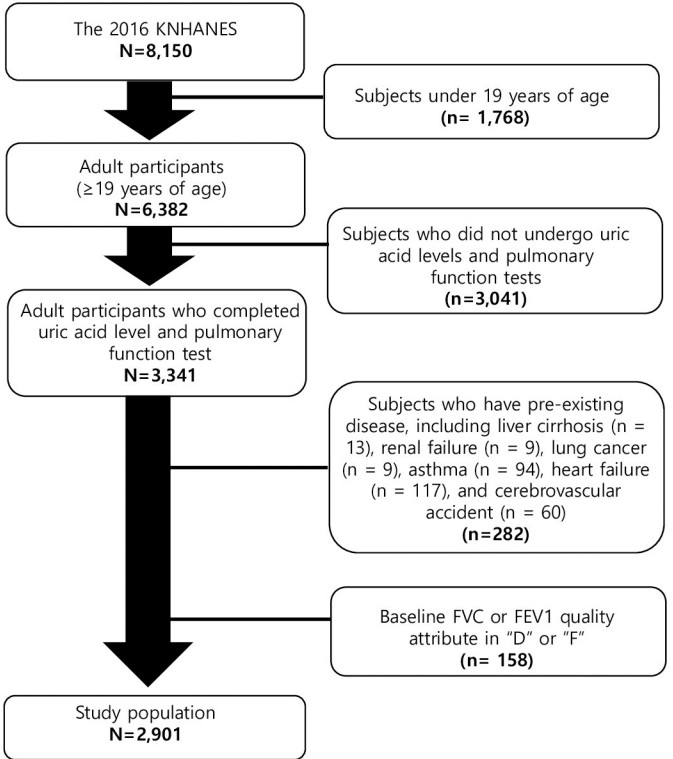

**Fig 1. Participants disposition to be included in this study.**

Men: Q1 <81.8%; Q2 81.9–89.9%; Q3 90.0–97.6%; Q4 ≥ 97.7%

Women: Q1 <84.7%; Q2 84.8–92.7%; Q3 92.8–100.0%; Q4 ≥ 100.1%

Obstructive pattern was defined as FEV1/FVC<0.70, and restrictive pattern was defined as FEV1/FVC≥0.7 and FVC <80%, While normal lung function was defined as FEV1/FVC≥0.70 and FVC ≥80% [22, 23].

## Laboratory methods

Serum uric acid levels were measured using a uricase colorimetry method with an autoanalyzer (7600-210; Hitachi, Tokyo, Japan), with serum uric acid levels > 7.0 mg/dL in men and > 6.0 mg/dL in women defined as hyperuricemia.

The estimated glomerular filtration rate (eGFR) was calculated based on serum creatinine levels using the Chronic Kidney Disease Epidemiology Collaboration equation. Renal impairment was defined as an eGFR < 60 mL/min/1.73 m$^2$ [24].

## Ethics statement

This study was approved by the institutional review board of Ilsan Paik Hospital, Republic of Korea (IRB 2019-10-012). After obtaining approval for the study protocol, the KNHANES dataset was made available at the request of the investigator. Because the dataset did not include any personal information and participants' consent had already been given for the KNHANES, our study was exempt from the requirement for informed consent.

## Statistical analyses

Statistical analyses were performed using SPSS for Windows software (ver. 21.0; SPSS Inc., Chicago, IL, USA). Demographic and clinical characteristics, including gender, age, education level (elementary school/junior high school/senior high school/college graduated), smoking status (non-/ex-/current smoker), alcohol consumption (none/1-3/week/≥4/week), aerobic exercise, obesity, hypertension, diabetes, and hyperuricemia were analyzed using chi-squared tests. Household income, FVC% predicted, FEV1% predicted, and serum uric acid were analyzed by independent samples *t*-tests. Age, education level, household income, smoking status, alcohol consumption, aerobic exercise, obesity, hypertension, diabetes, renal impairment, serum uric acid, and hyperuricemia of participants in the lowest quartile of FEV1 or FVC% predicted (Q1 group) were compared to those in the remaining quartiles (Q2–4 group) using independent samples *t*-tests or chi-squared tests. Comparisons of serum uric acid and the prevalence of hyperuricemia between Q1 group and Q2–4 group were done after adjusting for confounding factors in model 1 (Age) and 2 (all variables) (ANCOVA test). Serum uric acid and the prevalence of hyperuricemia were adjusted for confounding factors in model 1 (Age) and 2 (all variables). Spearman's correlation was used in order to determine the association between serum uric acid level and FVC% predicted, FEV1% predicted or FEV1/FVC. Age was adjusted in Model 1. In Model 2, age, education, household income (KRW), smoking (non-, ex-, and current smoker), alcohol consumption (none, 1-3/week, and ≥ 4/week), aerobic exercise, BMI (kg/m2), systolic BP (mmHg), fasting plasma glucose (mg/dL), and eGFR (mL/min/1.73 m2) were adjusted.

Multivariable logistic regression analyses were used to calculate the odds ratio (OR) of hyperuricemia for participants in the lowest quartile of FVC% predicted and FEV1% predicted, with age, education, household income, alcohol consumption, smoking, aerobic exercise, obesity, hypertension, diabetes, and renal impairment serving as covariates. All tests were two sided, with $p < 0.05$ considered indicative of statistical significance.

## Results

### Clinical characteristics of the study population

The demographic and clinical characteristics of the study population according to gender are shown in Table 1. The median age was 57 years (range: 40–80 years), and 57% of the participants were female. The mean FVC% predicted and FEV1% predicted in the general population were 91.1 ± 0.2 and 91.2 ± 0.2, respectively. The mean serum uric acid level was 4.9 ± 0.1 mg/dL, with an overall prevalence of hyperuricemia of 9.7%.

Men were found to have higher levels of education and household income, and were also more likely to smoke and engage in heavy alcohol consumption compared to women. The rates of aerobic exercise, obesity, hypertension, and diabetes were also higher in men than women. The mean serum uric acid level was 5.7 ± 0.1 mg/dL in men and 4.4 ± 0.1 mg/dL in women. The overall prevalence of hyperuricemia was 14.8% in men and 5.7% in women. Both the mean FVC% predicted and mean FEV1% predicted were higher in women than in men (89.6 ± 0.3 vs. 92.1 ± 0.3; $p < 0.001$, and 89.0 ± 0.4 vs. 92.9 ± 0.3; $p < 0.001$, respectively).

### Serum uric acid and the prevalence of hyperuricemia according to the FEV1/FVC 0.7

Overall, 74.7% of participants showed normal spirometric values. 12.1% and 13.2% of subjects have restrictive pattern and obstructive pattern in pulmonary function test, respectively. We compared the values of serum uric acid and prevalence of hyperuricemia in those with FEV1/

**Table 1. Clinical characteristics of study population.**

|  | Men (n = 1,266) | Women (n = 1,635) | p | Total (n = 2,901) |
|---|---|---|---|---|
| Age (years) | 56 (40–80) | 57 (40–80) | 0.870 | 57 (40–80) |
| Education (%) |  |  | <0.001 |  |
| Elementary school graduated | 20.3 | 32.2 |  | 27.0 |
| Junior high school graduated | 12.1 | 14.0 |  | 13.2 |
| Senior high school graduated | 30.6 | 31.3 |  | 31.0 |
| College graduated | 37.0 | 22.5 |  | 28.8 |
| Household Income (x10,000 KRW) | 423.0 ± 8.8 | 392.3 ± 7.9 | 0.009 | 405.7 ± 5.9 |
| Smoking (%) |  |  | <0.001 |  |
| Non-smoker | 19.1 | 92.3 |  | 60.4 |
| Ex- smoker | 49.0 | 3.0 |  | 23.1 |
| Current smoker | 31.9 | 4.7 |  | 16.6 |
| Alcohol consumption (%) |  |  | <0.001 |  |
| None | 19.7 | 40.0 |  | 31.2 |
| 1-3/week | 65.1 | 57.7 |  | 60.9 |
| ≥ 4/week | 15.2 | 2.3 |  | 7.9 |
| Aerobic exercise (%) | 45.7 | 40.4 | 0.005 | 42.7 |
| Obesity (%) | 41.4 | 36.8 | 0.011 | 38.8 |
| Hypertension (%) | 43.6 | 35.4 | <0.001 | 39.0 |
| Diabetes (%) | 17.5 | 12.7 | <0.001 | 14.8 |
| Renal impairment (%) | 4.3 | 3.5 | 0.245 | 3.9 |
| FVC% predicted | 89.6 ± 0.3 | 92.1 ± 0.3 | <0.001 | 91.1 ± 0.2 |
| FEV1% predicted | 89.0 ± 0.4 | 92.9 ± 0.3 | <0.001 | 91.2 ± 0.2 |
| Serum uric acid (mg/dL) | 5.7 ± 0.1 | 4.4 ± 0.1 | <0.001 | 4.9 ± 0.1 |
| Hyperuricemia (%) | 14.8 | 5.7 | <0.001 | 9.7 |

Data are expressed as mean ± SEM or %, except age. Age are expressed as median (minimum-maximum).

FVC, forced vital capacity; FEV1, forced expiratory volume in 1 second.

FVC ≥ 0.7 to those with < 0.7. The prevalence of hyperuricemia in those with FEV1/FVC ≥ 0.7 was 14.2% in men and 5.6% in women. The prevalence of hyperuricemia in those with FEV1/FVC < 0.7 was 17.2% in men and 8.6% in women. There is no significant difference in the prevalence of hyperuricemia according to the obstructive pattern in pulmonary function test, using FEV1/FVC 0.7 criteria. The mean serum uric acid level in those with FEV1/FVC ≥ 0.7 was 5.7 ± 0.1 mg/dL in men and 4.4 ± 0.1 mg/dL in women. The mean serum uric acid level in those with FEV1/FVC < 0.7 was 5.8 ± 0.1 mg/dL in men and 4.4 ± 0.1 mg/dL in women. There is also no significant difference in the mean serum uric acid levels according to the obstructive pattern in pulmonary function test.

## Clinical characteristics according to gender and FVC % predicted

We compared clinical characteristics between the participants in the lowest FVC% predicted quartile (Q1 group) and those in the remaining quartiles (Q2–4 group) (Table 2). In men, the median FVC% predicted of the Q1 and Q2–4 groups were 77.1 and 93.4, respectively. The Q2–4 group was younger, and had a higher education level and higher household income, compared to the Q1 group. The rates of obesity, hypertension, diabetes, and renal impairment were higher in the Q1 group. Serum uric acid levels, and the prevalence of hyperuricemia, were not statistically different between groups.

**Table 2. Clinical characteristics according to gender and FVC% predicted.**

| | Men (n = 1,266) | | | Women (n = 1,635) | | |
|---|---|---|---|---|---|---|
| | FVC% predicted Q1 (n = 313) | FVC% predicted Q2-4 (n = 953) | p | FVC% predicted Q1 (n = 406) | FVC% predicted Q2-4 (n = 1,229) | p |
| FVC% predicted | 77.1 (33.1–82.3) | 93.4 (82.5–121.6) | <0.001 | 79.4 (48.2–84.1) | 95.8 (84.1–138.6) | <0.001 |
| Age (years) | 64 (40–80) | 55 (40–80) | **<0.001** | 61 (40–80) | 55 (40–80) | **<0.001** |
| Education (%) | | | **0.001** | | | **<0.001** |
| Elementary school graduated | 26.5 | 18.3 | | 40.6 | 29.5 | |
| Junior high school graduated | 14.4 | 11.3 | | 14.5 | 13.8 | |
| Senior high school graduated | 29.1 | 31.2 | | 28.6 | 32.1 | |
| College graduated | 30.0 | 39.2 | | 16.3 | 24.6 | |
| Household Income (x10,000 KRW) | 353.2 ± 17.8 | 446.0 ± 10.0 | <0.001 | 348.2 ± 15.1 | 406.7 ± 9.2 | 0.001 |
| Smoking (%) | | | 0.165 | | | 0.523 |
| Non-smoker | 19.2 | 19.1 | | 93.1 | 92.0 | |
| Ex- smoker | 53.0 | 47.6 | | 3.2 | 2.9 | |
| Current smoker | 27.8 | 33.3 | | 3.7 | 5.0 | |
| Alcohol consumption (%) | | | **0.003** | | | **0.004** |
| None | 26.2 | 17.6 | | 46.8 | 37.8 | |
| 1-3/week | 58.1 | 67.4 | | 50.7 | 60.0 | |
| ≥ 4/week | 15.7 | 15.0 | | 2.5 | 2.2 | |
| Aerobic exercise (%) | 44.1 | 46.3 | 0.514 | 35.0 | 42.2 | **0.010** |
| Obesity (%) | 52.1 | 37.9 | <0.001 | 50.7 | 32.1 | <0.001 |
| Hypertension (%) | 56.5 | 39.3 | <0.001 | 48.3 | 31.2 | <0.001 |
| Diabetes (%) | 28.4 | 13.9 | <0.001 | 20.2 | 10.3 | <0.001 |
| Renal impairment (%) | 7.0 | 3.5 | **0.010** | 6.2 | 2.6 | **0.002** |
| Serum uric acid (mg/dL) | | | | | | |
| Unadjusted | 5.7 (5.6–5.9) | 5.7 (5.6–5.8) | 0.645 | 4.6 (4.5–4.7) | 4.3 (4.2–4.3) | <0.001 |
| Model 1 | 5.8 (5.6–5.9) | 5.7 (5.6–5.8) | 0.175 | 4.5 (4.5–4.6) | 4.3 (4.2–4.4) | <0.001 |
| Model 2 | 5.8 (5.6–5.9) | 5.7 (5.6–5.8) | 0.159 | 4.5 (4.4–4.6) | 4.3 (4.3–4.4) | **0.006** |
| Hyperuricemia (%) | | | | | | |
| Unadjusted | 16.9 | 14.2 | 0.233 | 9.9 | 4.4 | <0.001 |
| Model 1 | 18.0 (14.0–22.1) | 13.8 (11.5–16.2) | 0.077 | 9.1 (6.8–11.4) | 4.6 (3.4–5.9) | **0.001** |
| Model 2 | 17.6 (13.5–21.6) | 14.0 (11.7–16.2) | 0.136 | 7.9 (5.7–10.1) | 5.0 (3.8–6.2) | **0.025** |

Data are expressed as mean ± SEM or %, except FVC% predicted, age, serum uric acid, and hyperuricemia. FVC% predicted and age are expressed as median (minimum-maximum). Serum uric acid, and hyperuricemia are expressed as mean (95% CI). FVC, forced vital capacity.

Model 1: adjusted for age Model 2: adjusted for all variables

In women, the median FVC% predicted of the Q1 and Q2–4 groups were 79.4 and 95.8, respectively. The Q2–4 group was younger, had a higher education level and household income, and was more likely to engage in aerobic exercise compared to the Q1 group. The rates of frequent alcohol consumption, obesity, hypertension, diabetes, and renal impairment were all higher in the Q1 group. Serum uric acid levels (4.6 mg/dL vs. 4.3 mg/dL) and the prevalence of hyperuricemia (9.9% vs. 4.4%) were higher in the Q1 group compared to the Q2–4 group. These differences remained statistically significant after adjusting for all variables (model 1 and 2).

## Clinical characteristics according to gender and FEV1% predicted

Next, we compared clinical characteristics between participants in the lowest quartile of FEV1% predicted (Q1 group) and those in the remaining quartiles (Q2–4 group) (Table 3).

**Table 3. Clinical characteristics according to gender and FEV1% predicted.**

| | Men (n = 1,266) | | | Women (n = 1,635) | | |
|---|---|---|---|---|---|---|
| | FEV1% predicted Q1 (n = 315) | FEV1% predicted Q2-4 (n = 951) | p | FEV1% predicted Q1 (n = 411) | FEV1% predicted Q2-4 (n = 1,224) | p |
| FEV1% predicted | 74.9 (28.1–81.7) | 93.5 (81.8–129.8) | <0.001 | 78.9 (47.0–84.7) | 96.9 (84.7–158.0) | <0.001 |
| Age (years) | 62 (40–80) | 55 (40–80) | <0.001 | 57 (40–80) | 56 (40–80) | 0.044 |
| Education (%) | | | <0.001 | | | 0.626 |
| Elementary school graduated | 27.9 | 17.8 | | 33.1 | 31.9 | |
| Junior high school graduated | 16.8 | 10.5 | | 12.4 | 14.5 | |
| Senior high school graduated | 27.0 | 31.9 | | 32.8 | 30.7 | |
| College graduated | 28.3 | 39.9 | | 21.7 | 22.8 | |
| Household Income (x10,000 KRW) | 375.8 ± 18.1 | 438.7 ± 10.0 | 0.002 | 386.2 ± 15.3 | 394.3 ± 9.2 | 0.651 |
| Smoking (%) | | | 0.103 | | | 0.734 |
| Non- smoker | 15.2 | 20.4 | | 93.2 | 92.0 | |
| Ex-smoker | 49.8 | 48.7 | | 2.7 | 3.1 | |
| Current smoker | 34.9 | 30.9 | | 4.1 | 4.9 | |
| Alcohol consumption (%) | | | <0.001 | | | 0.019 |
| None | 24.4 | 18.2 | | 45.7 | 38.1 | |
| 1-3/week | 53.7 | 68.9 | | 51.8 | 59.7 | |
| ≥ 4/week | 21.9 | 12.9 | | 2.4 | 2.2 | |
| Aerobic exercise (%) | 42.2 | 46.9 | 0.152 | 41.1 | 40.2 | 0.772 |
| Obesity (%) | 41.0 | 41.5 | 0.895 | 36.0 | 37.0 | 0.723 |
| Hypertension (%) | 51.1 | 41.1 | 0.002 | 37.7 | 34.6 | 0.283 |
| Diabetes (%) | 24.8 | 15.0 | <0.001 | 14.8 | 12.0 | 0.146 |
| Renal impairment (%) | 5.4 | 4.0 | 0.298 | 4.6 | 3.1 | 0.162 |
| Serum uric acid (mg/dL) | | | | | | |
| Unadjusted | 5.7 (5.6–5.9) | 5.7 (5.6–5.8) | 0.508 | 4.4 (4.3–4.5) | 4.3 (4.3–4.4) | 0.095 |
| Model 1 | 5.8 (5.6–5.9) | 5.7 (5.6–5.8) | 0.212 | 4.4 (4.3–4.5) | 4.3 (4.3–4.4) | 0.158 |
| Model 2 | 5.8 (5.7–6.0) | 5.7 (5.6–5.8) | 0.105 | 4.4 (4.3–4.5) | 4.3 (4.3–4.4) | 0.226 |
| Hyperuricemia (%) | | | | | | |
| Unadjusted | 18.4 | 13.7 | 0.040 | 8.3 | 4.9 | 0.011 |
| Model 1 | 19.1 (15.2–23.1) | 13.4 (11.2–15.7) | 0.015 | 8.1 (5.8–10.3) | 5.0 (3.7–6.3) | 0.020 |
| Model 2 | 19.2 (15.3–23.2) | 13.4 (11.2–15.7) | 0.014 | 7.8 (5.6–10.0) | 5.0 (3.7–6.3) | 0.034 |

Data are expressed as mean ± SEM or %, except FEV1% predicted, age, serum uric acid, and hyperuricemia. FEV1% predicted and age are expressed as median (minimum-maximum). Serum uric acid, and hyperuricemia are expressed as mean (95% CI). FEV1, forced expiratory volume in 1 second.

Model 1: adjusted for age Model 2: adjusted for age, education, household income, smoking, alcohol consumption, hypertension, and diabetes

In men, the median FEV1% predicted of the Q1 and Q2–4 groups were 74.9 and 93.5, respectively. The Q2–4 group was younger age, and had higher education household income levels compared to the Q1 group. The rates of frequent alcohol consumption, hypertension, and diabetes were all higher in the Q1 group. Serum uric acid levels were not statistically different between groups (5.7 mg/dL vs. 5.7 mg/dL for the Q1 and Q2–4 groups, respectively). However, the prevalence of hyperuricemia was significantly higher in the Q1 group, even after adjusting for age, education, household income, smoking, alcohol consumption, hypertension, and diabetes (model 1 and 2).

In women, the median FEV1% predicted of the Q1 and Q2–4 groups were 78.9 and 96.9, respectively. The Q2–4 group was younger and consumed less alcohol compared to the Q1 group. Serum uric acid levels were not significantly different between the groups (4.4 mg/dL

vs. 4.3 mg/dL for the Q1 and Q2–4 groups, respectively). The prevalence of hyperuricemia (8.3% vs. 4.9%) was significantly higher in the Q1 group, even after adjusting for above variables (model 1 and 2).

## Correlations of serum uric acid with FVC% predicted, FEV1% predicted, and FEV1/FVC

No association was observed between serum uric acid levels and spirometric pulmonary function in men (Table 4). In contrast, unadjusted or adjusted FVC% predicted was negatively correlated with serum uric acid levels in women (Spearman correlation coefficient -0.145, $p < 0.001$). Unadjusted or adjusted FEV1% predicted was also negatively associated with serum uric acid levels (Spearman correlation coefficient -0.058, $p = 0.019$) in women, irrespective of age. There is no significant association between serum uric acid levels and FEV1/FVC (Table 4).

## Multivariable logistic regression analysis of the lowest quartile of FVC% predicted

Logistic regression analyses were performed to identify factors independently associated with the lowest quartile of FVC% predicted. Age, education level, household income, alcohol consumption, smoking, aerobic exercise, obesity, hypertension, diabetes, renal impairment, and hyperuricemia were included as potential confounders (Table 5).

Older age, obesity, and diabetes were associated with the lowest quartile of FVC% predicted for both men and women. In only women, hyperuricemia (OR 1.71, 95% CI 1.06–2.75, $p = 0.027$) was associated with lowest quartile of FVC% predicted.

## Multivariable logistic regression analyses of lowest quartile of FEV1% predicted

Logistic regression analyses were performed to identify factors independently associated with the lowest quartile of FEV1% predicted. Age, education level, household income, alcohol consumption, smoking, aerobic exercise, obesity, hypertension, diabetes, renal impairment, and hyperuricemia were included as potential confounders (Table 6). In men, age, current smoker, diabetes, and hyperuricemia (OR 1.54, 95% CI 1.07–2.22, $p = 0.021$) were associated with the lowest quartile of FEV1% predicted. In women, age and hyperuricemia (OR 1.70, 95% CI 1.06–2.74, $p = 0.028$) were associated with the lowest quartile of FEV1% predicted.

## Odds ratio of hyperuricemia for the lowest quartile of FCV% predicted and the lowest quartile of FEV1% predicted according to age $\geq$ 56 years old and $<$ 56 years old

Additionally, we showed odds ratios of hyperuricemia for the lowest quartile of FCV% predicted and the lowest quartile of FEV1% predicted according to age $\geq$ 56 years old and $<$ 56 years old (median age) in Table 7. In women, age $\geq$ 56 years old with hyperuricemia was associated with lowest quartile of FVC% predicted (OR 1.85, 95% CI 1.04–3.28, $p = 0.037$) and FEV1% predicted (OR 1.99, 95% CI 1.11–3.75, $p = 0.021$), respectively. In men, age $\geq$ 56 years old with hyperuricemia was associated with lowest quartile of FEV1% predicted (OR 1.75, 95% CI 1.05–2.94, $p = 0.033$), not FCV% predicted. Neither man nor woman younger than 56 years old was associated with lowest quartile of FCV% predicted or FEV1% predicted.

**Table 4. Correlation between serum uric acid and FVC% predicted or FEV1% predicted or FEV1/FVC.**

| Men | | r | p |
|---|---|---|---|
| FVC% predicted | | -0.028 | 0.323 |
| FVC% predicted model 1 | | -0.053 | 0.059 |
| FVC% predicted model 2 | | | |
| | Total | -0.059 | 0.037 |
| | Age < 56 years old | -0.059 | 0.153 |
| | Age ≥ 56 years old | -0.058 | 0.140 |
| FEV1% predicted | | -0.022 | 0.424 |
| FEV1% predicted model 1 | | -0.032 | 0.259 |
| FEV1% predicted model 2 | | | |
| | Total | -0.053 | 0.063 |
| | Age < 56 years old | -0.036 | 0.387 |
| | Age ≥ 56 years old | -0.067 | 0.087 |
| FEV1/FVC | | -0.029 | 0.306 |
| FEV1/FVC model 1 | | -0.019 | 0.510 |
| FEV1/FVC model 2 | | | |
| | Total | -0.009 | 0.740 |
| | Age < 56 years old | -0.034 | 0.407 |
| | Age ≥ 56 years old | -0.030 | 0.445 |
| Women | | r | p |
| FVC% predicted | | -0.145 | **<0.001** |
| FVC% predicted model 1 | | -0.122 | **<0.001** |
| FVC% predicted model 2 | | | |
| | Total | -0.095 | **<0.001** |
| | Age < 56 years old | -0.100 | **0.006** |
| | Age ≥ 56 years old | -0.087 | **0.011** |
| FEV1% predicted | | -0.058 | **0.019** |
| FEV1% predicted model 1 | | -0.064 | **0.010** |
| FEV1% predicted model 2 | | | |
| | Total | -0.092 | **<0.001** |
| | Age < 56 years old | -0.078 | **0.032** |
| | Age ≥ 56 years old | -0.099 | **0.004** |
| FEV1/FVC | | -0.018 | 0.476 |
| FEV1/FVC model 1 | | -0.029 | 0.240 |
| FEV1/FVC model 2 | | | |
| | Total | -0.015 | 0.556 |
| | Age < 56 years old | -0.026 | 0.481 |
| | Age ≥ 56 years old | -0.047 | 0.171 |

r, Spearman correlation coefficient. FVC, forced vital capacity; FEV1, forced expiratory volume in 1 second.

Model 1: adjusted for age

Model 2: adjusted for age, education, household income (KRW), smoking (non-, ex-, and current smoker), alcohol consumption (none, 1-3/week, and ≥ 4/week), aerobic exercise, BMI ($kg/m^2$), systolic BP (mmHg), fasting plasma glucose (mg/dL), and eGFR (mL/min/1.73 $m^2$)

**Table 5. Multivariable logistic regression analyses for lowest quartile of FVC% predicted.**

| | | Men | | Women | |
|---|---|---|---|---|---|
| | | Odds ratio (95% CI) | p | Odds ratio (95% CI) | p |
| Age (years) | 19–44 | Reference | | Reference | |
| | 45–64 | 1.98 (1.19–3.32) | **0.009** | 1.29 (0.86–1.96) | 0.228 |
| | 65- | 4.71 (2.67–8.33) | **<0.001** | 2.70 (1.63–4.46) | **<0.001** |
| Education | | | 0.907 | | 0.704 |
| | Elementary school graduated | Reference | | Reference | |
| | Junior high school graduated | 1.04 (0.65–1.57) | 0.857 | 1.10 (0.75–1.60) | 0.640 |
| | Senior high school graduated | 0.98 (0.66–1.45) | 0.908 | 1.19 (0.85–1.68) | 0.314 |
| | College graduated | 1.03 (0.67–1.57) | 0.904 | 1.07 (0.71–1.61) | 0.744 |
| Household income | | | 0.061 | | |
| | Quartile 1 (the lowest quartile) | Reference | | Reference | |
| | Quartile 2 | 0.71 (0.47–1.06) | 0.093 | 1.31 (0.93–1.86) | 0.128 |
| | Quartile 3 | 0.62 (0.40–0.95) | **0.028** | 1.48 (1.02–2.14) | **0.041** |
| | Quartile 4 (the highest quartile) | 0.63 (0.40–0.98) | **0.042** | 1.14 (0.76–1.69) | 0.526 |
| Alcohol consumption | | | 0.392 | | |
| | None | Reference | | Reference | |
| | 1-3/week | 0.81 (0.57–1.15) | 0.238 | 0.87 (0.68–1.12) | 0.283 |
| | ≥ 4/week | 0.74 (0.47–1.18) | 0.201 | 1.05 (0.48–2.27) | 0.910 |
| Smoking | | | | | |
| | Non-smoker | Reference | | Reference | |
| | Ex-smoker | 1.03 (0.71–1.50) | 0.879 | 1.09 (0.55–2.14) | 0.805 |
| | Current smoker | 1.08 (0.71–1.64) | 0.722 | 0.77 (0.42–1.40) | 0.385 |
| Aerobic exercise | | 1.03 (0.78–1.36) | 0.843 | 0.82 (0.64–1.05) | 0.124 |
| Obesity | | 1.94 (1.46–2.57) | **<0.001** | 1.74 (1.36–2.22) | **<0.001** |
| Hypertension | | 1.42 (1.07–1.89) | **0.016** | 1.29 (0.99–1.68) | 0.065 |
| Diabetes | | 1.75 (1.25–2.45) | **0.001** | 1.58 (1.14–2.21) | **0.007** |
| Renal impairment | | 0.84 (0.45–1.58) | 0.589 | 0.95 (0.52–1.75) | 0.867 |
| Hyperuricemia | | 1.32 (0.90–1.93) | 0.151 | 1.71 (1.06–2.75) | **0.027** |

FVC, forced vital capacity.

## Discussion

Using data from the 2016 KNHANES, the prevalence of hyperuricemia in the representative sample of Korean adults included in this study was 9.7% (14.8% in men, 5.7% in women). We found that hyperuricemia was inversely associated with spirometric pulmonary function in Korean population. This reverse correlation between hyperuricemia and pulmonary function was more pronounced in women and older age (≥56 years old).

Previous epidemiological studies have investigated the association between serum uric acid and pulmonary function in the general population, however, the results have been inconclusive. A recent Japanese study showed similar results to those presented here, including significant inverse correlations of serum uric acid levels with spirometric parameters in female subjects who participated in an annual health check [15].

In contrast, a second study found a significant positive correlation between pulmonary function and serum uric acid levels, in both males and females recruited from Kangbuk Samsung Hospital Health Screening Center, Seoul, Korea [16]. While that study was similar to the analysis presented herein, in that it used a cross-sectional design, there are some significant

**Table 6. Multivariable logistic regression analyses for lowest quartile of FEV1% predicted.**

| | | Men | | Women | |
|---|---|---|---|---|---|
| | | Odds ratio (95% CI) | p | Odds ratio (95% CI) | p |
| Age (years) | 19–44 | Reference | | Reference | |
| | 45–64 | 1.32 (0.83–2.09) | 0.240 | 1.50 (1.02–2.21) | **0.039** |
| | 65- | 2.26 (1.33–3.83) | 0.002 | 1.74 (1.07–2.83) | **0.025** |
| Education | | | | | |
| | Elementary school graduated | Reference | | Reference | |
| | Junior high school graduated | 1.10 (0.71–1.71) | 0.670 | 0.90 (0.61–1.32) | 0.584 |
| | Senior high school graduated | 0.68 (0.46–0.99) | **0.046** | 1.22 (0.87–1.70) | 0.249 |
| | College graduated | 0.70 (0.46–1.05) | 0.087 | 1.15 (0.78–1.69) | 0.486 |
| Household income | | | | | |
| | Quartile 1 (the lowest quartile) | Reference | | Reference | |
| | Quartile 2 | 0.72 (0.48–1.07) | 0.099 | 1.09 (0.76–1.54) | 0.650 |
| | Quartile 3 | 0.70 (0.46–1.07) | 0.100 | 1.23 (0.85–1.78) | 0.270 |
| | Quartile 4 (the highest quartile) | 0.79 (0.51–1.23) | 0.298 | 1.14 (0.78–1.67) | 0.510 |
| Alcohol consumption | | | | | |
| | None | Reference | | Reference | |
| | 1-3/week | 0.71 (0.50–1.00) | 0.051 | 0.74 (0.58–0.94) | 0.014 |
| | ≥ 4/week | 1.23 (0.80–1.89) | 0.353 | 0.98 (0.46–2.08) | 0.958 |
| Smoking | | | | | |
| | Non-smoker | Reference | | Reference | |
| | Ex-smoker | 1.35 (0.92–1.98) | 0.127 | 0.96 (0.48–1.92) | 0.909 |
| | Current smoker | 1.80 (1.19–2.72) | **0.006** | 0.90 (0.51–1.60) | 0.728 |
| Aerobic exercise | | 0.95 (0.72–1.24) | 0.680 | 1.04 (0.82–1.31) | 0.742 |
| Obesity | | 1.00 (0.76–1.32) | 0.984 | 0.89 (0.69–1.14) | 0.346 |
| Hypertension | | 1.14 (0.86–1.51) | 0.356 | 1.02 (0.78–1.33) | 0.886 |
| Diabetes | | 1.50 (1.07–2.10) | **0.018** | 1.22 (0.87–1.72) | 0.250 |
| Renal impairment | | 0.67 (0.351.28) | 0.222 | 1.04 (0.55–1.96) | 0.907 |
| Hyperuricemia | | 1.54 (1.07–2.22) | **0.021** | 1.70 (1.06–2.74) | **0.028** |

FEV1, forced expiratory volume in 1 second.

differences between the two studies. First, the subjects who participated in the Kangbuk Samsung Health Study were significantly younger than those included in our study (mean age were 40 and 56 years, respectively). Because age might be a moderator for the reverse

**Table 7. Odds ratio of hyperuricemia for the lowest quartile of FCV% predicted and FEV1% predicted according to age ≥ 56 years old and < 56 years old.**

| | Men (n = 1266) | | Women (n = 1635) | |
|---|---|---|---|---|
| | Odds ratio (95% CI) | p | Odds ratio (95% CI) | p |
| for lowest quartile of FCV% predicted | | | | |
| < 56 years old | 0.36 (0.77–2.43) (n = 606) | 0.291 | 1.54 (0.63–3.76) (n = 761) | 0.347 |
| ≥ 56 years old | 1.32 (0.79–2.22) (n = 660) | 0.293 | 1.85 (1.04–3.28) (n = 874) | **0.037** |
| for lowest quartile of FEV1% predicted | | | | |
| < 56 years old | 1.36 (0.79–2.33) | 0.269 | 1.20 (0.49–2.96) | 0.692 |
| ≥ 56 years old | 1.75 (1.05–2.94) | **0.033** | 1.99 (1.11–3.57) | **0.021** |

Covariates: age, education level, household income, smoking status, alcohol consumption, aerobic exercise, obesity, hypertension, diabetes, and renal impairment.
FVC, forced vital capacity; FEV1, forced expiratory volume in 1 second.

association between pulmonary function and serum uric acid, we performed sub-analysis about odds ratio of hyperuricemia for the lowest quartile of FCV% predicted and the lowest quartile of FEV1% predicted according to age ≥ 56 years old and < 56 years old. Actually, correlation between lowest quartile of FCV% and/or FEV1% predicted and hyperuricemia was apparent only in women ≥ 56 years old, not in women <56 years old. There was also significant association between hyperuricemia and lowest quartile of FEV1% predicted in men ≥ 56 years old, not in men <56 years old. Furthermore, the Kangbuk Samsung Health Study was performed in Seoul, where the higher average socioeconomic status of that region could have constituted a sampling bias. Using a definition of hyperuricemia of > 7.0 mg/dL uric acid in men and > 6.0 mg/dL in women, the prevalence rates of hyperuricemia in the Kangbuk Samsung Health study (25.5% in men and 8.5% in women) were much higher than those in both our study (14.8% in men and 5.7% in women) and a previous study of the Korean population (14.3% in men and 2.2% in women) [25]. The analytical methods were also different between the studies. We compared serum uric acid level and the prevalence of hyperuricemia between participants in the lowest quartile of pulmonary function and those in the other quartiles, while the Kangbuk Samsung Health Study compared pulmonary function between hyperuricemia and normouricemia groups. In this study, we included age, education level, household income, smoking status, alcohol consumption, aerobic exercise, obesity, hypertension, and diabetes as covariates; only some of these variables overlapped with those adjusted for in the Kangbuk Samsung Health Study, which included age, BMI, smoking status, alcohol consumption, liver and renal function, lipid profiles, C-reactive protein level, mean blood pressure, and glycosylated hemoglobin (HbA1c) level. Therefore, we believe that differences in study populations, potential confounders including age and sex, and analytical methods may be responsible for the discrepancies in results seen among otherwise similar cross-sectional studies.

In this study, there was sex-difference in the association between serum uric acid and low pulmonary function as well as age. No association was observed between serum uric acid levels and spirometric pulmonary function in men. In contrast, unadjusted or adjusted FVC% predicted and FEV1% predicted were negatively correlated with serum uric acid levels in women, irrespective of age. Regarding to hyperuricemia, in women, hyperuricemia was associated with both lowest quartile of FVC% predicted and FEV1 predicted. However, in men, hyperuricemia was associated with only the lowest quartile of FEV1% predicted, not FVC% predicted. These results suggested that reverse correlation between serum uric acid and pulmonary function was more pronounced in women. However, the mechanism of the age or sex-specific difference between them is still unclear. Previously, although several studies showed that sex differences in the association between hyperuricemia and cardiometabolic risk, there has been no studies about sex differences in the relation between hyperuricemia and lung function [26–29].

One possibility is that there is an inherent difference in uric acid metabolism between the sexes, including a higher fractional urate excretion and lower serum urate concentration in women compared to men [26, 30]. We speculated that estrogen change might play a role in mediation of the association between serum uric acid and lung function. Another possibility is that other cofactors influencing pulmonary function not considered in this study may also be affected by gender.

Oxidative stress is thought to play a mediating role in the association between uric acid level and impaired lung function. The final reaction necessary for uric acid production involves the conversion of xanthine to uric acid, which is catalyzed by the enzyme xanthine oxidoreductase (XOR) [31]. During this process, XOR uses molecular oxygen as an electron acceptor, producing superoxide anions and other ROS [32]. The lungs could be a major target organ for exogenous oxidants and endogenous ROS generated by inflammatory cells [33], with evidence of higher uric acid concentrations within the epithelial lining fluid of the airways [33, 34]. ROS

and cellular injury have also been implicated in a variety of pulmonary diseases, including asthma, COPD, acute respiratory distress syndrome (ARDS), and cystic fibrosis [33].

However, because high level of uric acid could be viewed as a 'double-edge sword'; it may have beneficial antioxidant effects, paradoxically, any potential association between hyperuricemia and lung function in a cross-sectional study must be interpreted carefully, putting confounding factors aside [35–37].

Other mechanisms, such as hypoxia and systemic inflammation, may also modulate the association between hyperuricemia and poor pulmonary function [38]. However, this study excluded subjects with overt clinical disease, such as asthma, lung cancer, or heart failure diagnosed by a doctor.

The major strength of this study was the inclusion of a large, representative sample of adult Koreans. To the best of our knowledge, this is the large scale study to demonstrate an inverse association between hyperuricemia and pulmonary function in general population using nationally representative data. However, our study also had some limitations. First, we did not take into account the use of uric acid-lowering drugs, which reduce serum uric acid levels. Second, we used a single measurement of uric acid. Third, although we adjusted for many confounding factors, there may be other potentially important variables that were not accounted for, similar to other cross-sectional studies. Finally, we could not establish a causal relationship between hyperuricemia and poor pulmonary function due to the inherent limitations of the retrospective design.

Lastly, we could not identify a definitive mechanism underlying the age- and sex-specific association between pulmonary function and hyperuricemia.

In conclusion, hyperuricemia was associated with lowest quartile of FEV1% or FVC% predicted using data taken from the 2016 KNHANES. This correlation between hyperuricemia and low pulmonary function was more pronounced in women and older age. Further epidemiologic studies will be required to confirm this reverse association between serum uric acid and pulmonary function.

## Author Contributions

**Conceptualization:** Dong-Jun Kim.

**Data curation:** Dong-Jun Kim.

**Formal analysis:** Dong-Jun Kim.

**Investigation:** Dong-Jun Kim.

**Methodology:** Dong-Jun Kim.

**Software:** Dong-Jun Kim.

**Supervision:** Dong-Jun Kim.

**Writing – original draft:** Jae Won Hong.

**Writing – review & editing:** Jung Hyun Noh.

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
