## [Decision Letter · Decision Letter 0]

7 May 2020

PONE-D-19-34352

Association between Uric Acid and Pulmonary Function in Korean Adults: The 2016 Korea National Health and Nutrition Examination Survey

PLOS ONE

Dear Dr. Kim,

Thank you for submitting your manuscript to PLOS ONE. After careful consideration, we feel that it has merit but does not fully meet PLOS ONE’s publication criteria as it currently stands. Therefore, we invite you to submit a revised version of the manuscript that addresses the points raised during the review process.

Specifically, the reviewers have raised overlapping concerns about the statistical methodology employed in the manuscript as well as the reporting of potential confounding variables in the analysis. 

We would appreciate receiving your revised manuscript by Jun 20 2020 11:59PM. To enhance the reproducibility of your results, we recommend that if applicable you deposit your laboratory protocols in protocols.io, where a protocol can be assigned its own identifier (DOI) such that it can be cited independently in the future. For instructions see: http://journals.plos.org/plosone/s/submission-guidelines#loc-laboratory-protocols

We look forward to receiving your revised manuscript.

Kind regards,

Richard Hodge

Academic Editor

PLOS ONE

2. We noticed you have some minor occurrence(s) of overlapping text with the following previous publication(s), which needs to be addressed:

https://doi.org/10.1016/j.cca.2018.05.046

https://doi.org/10.1371/journal.pone.0167210

In your revision ensure you cite all your sources (including your own works), and quote or rephrase any duplicated text outside the Methods section. Further consideration is dependent on these concerns being addressed.

4. In the ethics statement in the manuscript and in the online submission form, please provide additional information about the patient records/samples used in your retrospective study. Specifically, please ensure that you have discussed whether all data/samples were fully anonymized before you accessed them and/or whether the IRB or ethics committee waived the requirement for informed consent. If patients provided informed written consent to have data/samples from their medical records used in research, please include this information.

6. Please amend either the abstract on the online submission form (via Edit Submission) or the abstract in the manuscript so that they are identical.

Reviewers' comments:

Reviewer's Responses to Questions

**Comments to the Author**

1. Is the manuscript technically sound, and do the data support the conclusions?

Reviewer #1: Yes

Reviewer #2: Yes

Reviewer #3: Partly

2. Has the statistical analysis been performed appropriately and rigorously? 

Reviewer #1: Yes

Reviewer #2: Yes

Reviewer #3: Yes

3. Have the authors made all data underlying the findings in their manuscript fully available?

Reviewer #1: Yes

Reviewer #2: Yes

Reviewer #3: Yes

4. Is the manuscript presented in an intelligible fashion and written in standard English?

Reviewer #1: Yes

Reviewer #2: Yes

Reviewer #3: No

5. Review Comments to the Author

Reviewer #1: Dear Editor:

The authors made a cross-sectional study for an associaiton of serum uric acid and FVC predicted and FEV1 predicted in the general population in Korea. They found a sex difference in the association where hyperuricemia was associated poor FVC and FEV1 predicted in women only.

In general the paper is well writtern; however some major issues should be addressed.

1. In title, the authors used "pulmonary function" to represent FVC and FEV1 predicted seemed too broad covering. In addition, it is suggested to add serum preceding uric acid.

2. In the background of the abstract, it is no correct that there has been some papers regarding the assocaition between SUA and FVC and FEV1 predicted.

3. In methods: pulmonary function test, the authors should mention what the prevalence of COPD and compare the values of SUA and prevalence of hyperuricemia in those with FEV1/FVC <70% to those those >=70%

4. In statistical analyses, smoking status should be added in the logistic regression analysis in men and women. In addtion the authors did not describe the method for table 4. I also suggest to take a look on FEV1/FVC using the similar analyses.

5. The authors cited the ref 19 using a Korean cohort and suggested that age might be a moderator for the association. The authors should make a subcohort study for those <=56 years (mean age) and those > 56 years to compre the result of ref 19.

6. There were no discussions for the sex difference in the association between SUA and FVC and FEV1 predicted. Some papers have done for the sex difference for the SUA and metabolic diseases (e.g. Lin YK, et al. Sex-specific association of hyperuricemia with cardiometabolic abnormalities in a military cohort: The CHIEF study. Medicine (Baltimore). 2020 Mar;99(12):e19535.; and Lin JW, et al. Sex-Specific Association Between Serum Uric Acid and Elevated Alanine Aminotransferase in a Military Cohort: The CHIEF Study. Endocr Metab Immune Disord Drug Targets. 2019;19(3):333-340).

Reviewer #2: This manuscript aims to investigate the association between uric acid levels and pulmonary function in general Korean population who participated in the 2016 Korea National Health and Nutrition Examination Survey (KNHANES). The cross-sectional and nationally representative survey showed that hyperuricemia was associated with low pulmonary function in Korean women.

However, some questions need to be considered.

1. Overall, the design of this study does not allow determination of causality. Please be sure to eliminate any implication of causality in the manuscript.

2. Abstract: It is better to limit the number of words up to 300 in the abstract.

(1) The methods were too simple; the study outcomes should be described in the method section.

(2) The full name of the abbreviation should be presented for the first time to show in the manuscript. For example, the full name of FVC% predicted and FEV1% predicted.

(2) Result: The content of the result should be more brief and clear.

3. Introduction:

(1) Please provide the references of the previous studies about the association between serum uric acid levels and pulmonary function.

4. Method:

(1) Please add the reference of “obesity”.

(2) What sphygmomanometer was used should be described.

(3) Whether the final participants included participants with pulmonary disease (such as COPD, etc) should be described in detail.

(4) Please provide the flow chart of study participants.

(5) Why using the GFR rather than eGFR?

(6) Why not include smoking status in the logistic regression model? And why the analyses for FCV% or FEV1% was included different covariates in the model? How were the covariates of the adjusted model selected?

(7) Did other covariates (such as biochemical indicator) collected in the present study?

5. Result:

(1) Clinical characteristics of the study population: Paragraph 2, “compared to” may be better than “relative to”.

(2) The detail value of odds ratio and its 95% confidence interval of “1.68-fold” should be presented.

(3) In the Table 4, why not provide model 1, 2, 3 in the men?

(4) Univariate or multivariate logistic regression was used in the Table 5 and Table 6?

(5) The association between serum uric acid and lowest quartile of FVC% predicted or lowest quartile of FEV1% predicted should be analyzed.

(6) How about the stratified analyses of the association between uric acid and the FVC% predicted and FEV1% predicted?

6. Discussion:

(1) Paragraph 3: “mean age is” might better than “mean age =”.

(2) Paragraph 7: The description, “the first study” is inappropriate.

7. Overall, the quality of date analysis, and the paper writing should be further improved.

Reviewer #3: The authors proposed the association between hyperuricemia and low pulmonary function in women not in men. The idea is interesting. However, many detailed information was lacking, for example, about 40% people included in the final analysis, making it difficult to judge the validity of the study findings. My comments are:

1. In the current study, authors defined smoking category as non-smoker, ex-smoker and current smoker. Have authors considered to categorize as active smoker (defined as participants who reported smoking ≥100 cigarettes in their lifetime and who currently smoked) and Passive smokers (defined as those who did not smoke but had been exposed to tobacco smoke either in the workplace or at home during the last week)? If it is possible, it is strongly recommended to change current definition to refine their analysis. Authors also need to explain their definition of “non-smoker” and “ex-smoker” in current study. The “non-smokers” can be defined as participants who smoked <100 cigarettes during their lifetime, had not smoked currently, and had not been exposed to second hand tobacco smoke. The “ex-smokers” can be defined as participants who smoked >100 cigarettes during their lifetime, had not smoked currently, and had not been exposed to second hand tobacco smoke”. There are many references available that authors can refer it to their finding. Considering that smoking may have a long-term effect on SUA/hyperuricemia, the finding may be biased from low smoking rate in women.

2. Since gender is a strong factor for SUA/hyperuricemia, the authors need to explain their finding only in women. By doing so, the readers could get much clear picture of distribution of potential confounders.

3. By adjusting eGFR in model 2, the association became stronger. Authors need to explain this phenomenon with plausible mechanism in discussion.

4. Both alcohol consumption and BMI were strong risk factors for SUA/hyperuricemia. In the current analysis, residual confounding by these two factors may be one potential explanation for the current findings. Authors also need to address these issues with further analysis.

5. Authors need to address the method of multiple imputation in current analysis and how many percentages of covariates were imputed. If not, authors need to address how to deal with missing values.

6. PLOS authors have the option to publish the peer review history of their article (what does this mean?). If published, this will include your full peer review and any attached files.

Reviewer #1: Yes: Gen-Min Lin

Reviewer #2: No

Reviewer #3: No

---

## [Author Response · Author response to Decision Letter 0]

30 Jul 2020

Reviewer #1: Dear Editor:

The authors made a cross-sectional study for an associaiton of serum uric acid and FVC predicted and FEV1 predicted in the general population in Korea. They found a sex difference in the association where hyperuricemia was associated poor FVC and FEV1 predicted in women only.

In general the paper is well writtern; however some major issues should be addressed.

1. In title, the authors used "pulmonary function" to represent FVC and FEV1 predicted seemed too broad covering. In addition, it is suggested to add serum preceding uric acid.

Thank you for your comments.

We changed the title to “Association between Serum Uric Acid and Spirometric Pulmonary Function in Korean Adults: The 2016 Korea National Health and Nutrition Examination Survey’.

2. In the background of the abstract, it is no correct that there has been some papers regarding the assocaition between SUA and FVC and FEV1 predicted.

We modified the background of the abstract, as follows

A limited number of epidemiological studies have investigated the association between serum uric acid levels and pulmonary function in the general population, however, the results have been inconclusive.

3. In methods: pulmonary function test, the authors should mention what the prevalence of COPD and compare the values of SUA and prevalence of hyperuricemia in those with FEV1/FVC <70% to those those >=70%

Spirometric values

 Normal PFT Restrictive pattern Obstructive pattern 

men 65.3% 12.6% 22.0%

women 81.9% 11.7% 6.4%

total 74.7% 12.1% 13.2%

Serum uric acid and the prevalence of hyperuricemia according to the FEV1/FVC 0.7

 FEV1/FVC ≥ 0.7 FEV1/FVC < 0.7 p

Hyperuricemia(%) men 14.2 17.2 0.216

 women 5.6 8.6 0.193

Serum uric acid (mg/dl) men 5.7 ± 0.1 5.8 ± 0.1 0.196

 women 4.4 ± 0.1 4.4 ± 0.1 0.817

Mean ± SEM

Because all spirometry values were prebronchodilator results, we could not make an accurate diagnosis of COPD, directly. However, as you suggested, we described the prevalence of obstructive pattern and the values of serum uric acid and prevalence of hyperuricemia in those with FEV1/FVC ≥ 0.7 to those with < 0.7, in the method and result section, as follows,

Method section> Pulmonary function test

Obstructive pattern was defined as FEV1/FVC <0.70, and restrictive pattern was defined as FEV1/FVC≥ 0.7 and FVC <80%, While normal lung function was defined as FEV1/FVC≥0.70 and FVC ≥80% [22, 23].

Result section> serum uric acid and the prevalence of hyperuricemia according to the FEV1/FVC

Overall, 74.7% of participants showed normal spirometric values. 12.1 % and 13.2 % of subjects have restrictive pattern and obstructive pattern in pulmonary function test, respectively. We compared the values of serum uric acid and prevalence of hyperuricemia in those with FEV1/FVC ≥ 0.7 to those with < 0.7. The prevalence of hyperuricemia in those with FEV1/FVC ≥ 0.7 was 14.2% in men and 5.6% in women. The prevalence of hyperuricemia in those with FEV1/FVC <0.7 was 17.2% in men and 8.6% in women. There is no significant difference in the prevalence of hyperuricemia according to the obstructive pattern in pulmonary function test, using FEV1/FVC 0.7 criteria. The mean serum uric acid level in those with FEV1/FVC ≥ 0.7 was 5.7 ± 0.1 mg/dL in men and 4.4 ± 0.1 mg/dL in women. The mean serum uric acid level in those with FEV1/FVC < 0.7 was 5.8 ± 0.1 mg/dL in men and 4.4 ± 0.1 mg/dL in women. There is also no significant difference in the mean serum uric acid levels according to the obstructive pattern in pulmonary function test.

4. In statistical analyses, smoking status should be added in the logistic regression analysis in men and women. In addtion the authors did not describe the method for table 4. I also suggest to take a look on FEV1/FVC using the similar analyses.

Thank you for your comments. As you pointed out, we added smoking status in logistic regression analysis and modified Table 5 and 6. Additionally, as your suggestion, we examined the correlation between serum uric acid and FEV1/FVC and modified Table 4.

5. The authors cited the ref 19 using a Korean cohort and suggested that age might be a moderator for the association. The authors should make a subcohort study for those <=56 years (mean age) and those > 56 years to compre the result of ref 19.

We performed these sub-cohort study according to age and modified Table 4 and added Table 7.

We also added this content in Result section, as follows,

Result section>

“Correlations of serum uric acid with FVC% predicted, FEV1% predicted, and FEV1/FVC

No association was observed between serum uric acid levels and spirometric pulmonary function in men (Table 4). In contrast, unadjusted or adjusted FVC% predicted was negatively correlated with serum uric acid levels in women (Spearman correlation coefficient -0.145, p < 0.001). Unadjusted or adjusted FEV1% predicted was also negatively associated with serum uric acid levels (Spearman correlation coefficient -0.058, p = 0.019) in women, irrespective of age. There is no significant association between serum uric acid levels and FEV1/FVC (Table 4).”

“Odds ratio of hyperuricemia for the lowest quartile of FCV% predicted and the lowest quartile of FEV1% predicted according to age ≥ 56 years old and < 56 years old.

Additionally, we showed odds ratios of hyperuricemia for the lowest quartile of FCV% predicted and the lowest quartile of FEV1% predicted according to age ≥ 56 years old and < 56 years old (median age) in Table 7. In women, age ≥ 56 years old with hyperuricemia was associated with lowest quartile of FVC% predicted (OR 1.85, 95% CI 1.04-3.28, p = 0.037) and FEV1 % predicted (OR 1.99, 95% CI 1.11–3.75, p = 0.021), respectively. In men, age ≥ 56 years old with hyperuricemia was associated with lowest quartile of FEV1 % predicted (OR 1.75, 95% CI 1.05–2.94, p = 0.033), not FCV% predicted. Neither man nor woman younger than 56 years old was associated with lowest quartile of FCV% predicted or FEV1% predicted.”

We added this content in Discussion section, as follows,

“We found that hyperuricemia was inversely associated with spirometric pulmonary function in Korean population. This reverse correlation between hyperuricemia and pulmonary function was more pronounced in women and older age (≥56 years old).”

“Because age might be a moderator for the reverse association between pulmonary function and serum uric acid, we performed sub-analysis about odds ratio of hyperuricemia for the lowest quartile of FCV% predicted and the lowest quartile of FEV1% predicted according to age ≥ 56 years old and < 56 years old (median age). Actually, correlation between lowest quartile of FCV% and/or FEV1% predicted and hyperuricemia was apparent only in women ≥ 56 years old, not in women <56 years old. There was also significant association between hyperuricemia and lowest quartile of FEV1 % predicted in men ≥ 56 years old, not in men <56 years old.”

6. There were no discussions for the sex difference in the association between SUA and FVC and FEV1 predicted. Some papers have done for the sex difference for the SUA and metabolic diseases (e.g. Lin YK, et al. Sex-specific association of hyperuricemia with cardiometabolic abnormalities in a military cohort: The CHIEF study. Medicine (Baltimore). 2020 Mar;99(12):e19535.; and Lin JW, et al. Sex-Specific Association Between Serum Uric Acid and Elevated Alanine Aminotransferase in a Military Cohort: The CHIEF Study. Endocr Metab Immune Disord Drug Targets. 2019;19(3):333-340).

Thank you for your comments. We added discussion about the sex difference in the association between serum uric acid and low pulmonary function, as follows.

“In this study, there was sex-difference in the association between serum uric acid and low pulmonary function as well as age. No association was observed between serum uric acid levels and spirometric pulmonary function in men. In contrast, unadjusted or adjusted FVC% predicted and FEV1% predicted were negatively correlated with serum uric acid levels in women, irrespective of age. Regarding to hyperuricemia, in women, hyperuricemia was associated with both lowest quartile of FVC% predicted and FEV1 predicted. However, in men, hyperuricemia was associated with only the lowest quartile of FEV1% predicted, not FVC% predicted. These results suggested that reverse correlation between serum uric acid and pulmonary function was more pronounced in women.

One possibility is that there is an inherent difference in uric acid metabolism between the sexes, including a higher fractional urate excretion and lower serum urate concentration in women compared to men [24, 25]. Previously, although several studies showed that sex differences in the association between hyperuricemia and cardiometabolic risk, there has been no studies about sex differences in the relation between hyperuricemia and lung function [25-27]. Women may have greater xanthium oxidase activity and reactive oxygen species (ROS) production, resulting in severe vascular inflammation, arterial stiffness, and elevated blood pressure as compared with men. We speculated that estrogen change might play a role in mediation of the association between serum uric acid and lung function. Another possibility is that other cofactors influencing pulmonary function not considered in this study may also be affected by gender. 

Reviewer #2: This manuscript aims to investigate the association between uric acid levels and pulmonary function in general Korean population who participated in the 2016 Korea National Health and Nutrition Examination Survey (KNHANES). The cross-sectional and nationally representative survey showed that hyperuricemia was associated with low pulmonary function in Korean women.”

However, some questions need to be considered.

1. Overall, the design of this study does not allow determination of causality. Please be sure to eliminate any implication of causality in the manuscript.

Thank you for your comments. We eliminated all the “-fold higher risk” sentences in multivariable logistic regression analyses. 

2. Abstract: It is better to limit the number of words up to 300 in the abstract.

(1) The methods were too simple; the study outcomes should be described in the method section.

Thank you for your comments. We modified the methods section in Abstract, as follows,

“Among the 8,150 participants who participated in the 2016 Korea National Health and Nutrition Examination Survey, 2,901 participants were analyzed in this study. Subjects were divided into four groups according to forced vital capacity (FVC)% predicted or forced expiratory volume in 1 second (FEV1) % predicted quartiles. Participants in the lowest quartile of FVC % predicted and FEV1% predicted were compared to those in the remaining quartiles according to age, education level, household income, smoking status, alcohol consumption, aerobic exercise, obesity, hypertension, diabetes, renal impairment, serum uric acid, and hyperuricemia. Multivariable logistic regression analyses were used to calculate the odds ratio of hyperuricemia for participants in the lowest quartile of FVC% and FEV1 predicted, with above covariates.”

(2) The full name of the abbreviation should be presented for the first time to show in the manuscript. 

For example, the full name of FVC% predicted and FEV1% predicted.

Thank you for your comments.

We added the full name of the abbreviation in abstract, as follows,

forced vital capacity (FVC)

forced expiratory volume in 1 second (FEV1)

odds ratio (OR)

(2) Result: The content of the result should be more brief and clear.

As you suggested, we modified the result section in abstract, as follows,

In women, hyperuricemia was associated with lowest quartile of FVC% predicted (OR 1.71, 95% CI 1.06–2.75, p = 0.027) and FEV1 predicted (OR 1.70, 95% CI 1.06–2.74, p = 0.028) respectively, serving as above confounding variables. In men, hyperuricemia (OR 1.54, 95% CI 1.07–2.22, p = 0.021) was associated with the lowest quartile of FEV1% predicted, not FVC% predicted.

According to median age, in women, age≥ 56 years old with hyperuricemia was associated with lowest quartile of FVC% predicted (OR 1.85, 95% CI 1.04-3.28, p = 0.037) and FEV1 % predicted (OR 1.99, 95% CI 1.11–3.75, p = 0.021), respectively. In men, age≥ 56 years old with hyperuricemia was associated with lowest quartile of FEV1 % predicted (OR 1.75, 95% CI 1.05–2.94, p = 0.033), not FCV% predicted. 

3. Introduction:

(1) Please provide the references of the previous studies about the association between serum uric acid levels and pulmonary function.

Thank you for your comments.

We added references” A limited number of epidemiological studies have investigated the association between serum uric acid levels and pulmonary function in the general population, however, the results have been inconclusive [15,16].

4. Method:

(1) Please add the reference of “obesity”.

We added the reference “Obesity was defined as a body mass index (BMI) ≥25 kg/m2 according to the Asia-Pacific obesity classification.”

REFERENCE> Appropriate body-mass index for Asian populations and its implications for policy and intervention strategies. Lancet. 2004;363(9403):157-63. Epub 2004/01/17. doi: 10.1016/s0140-6736(03)15268-3. PubMed PMID: 14726171.

(2) What sphygmomanometer was used should be described.

Systolic and diastolic blood pressure were measured by standard methods using a standard mercury sphygmomanometer (Baumanometer, WA Baum Co. Inc., Copiague, NY, USA).

 (3) Whether the final participants included participants with pulmonary disease (such as COPD, etc) should be described in detail.

In the health interview about medical history, there was no survey item on the presence of “COPD”, but “asthma” on questionnaire.

A total of 282 subjects were excluded from the analysis due to pre-existing diseases, including liver cirrhosis (n = 13), renal failure (n = 9), lung cancer (n = 9), asthma (n = 94), heart failure (n = 117), and cerebrovascular accident (n = 60), for a final total of 3,059 subjects. 

(4) Please provide the flow chart of study participants. 

Thank you for your comment.

As you suggested, we added flow chart of study participants in Figure1. 

(5) Why using the GFR rather than eGFR?

We used estimated GFR using CKD-EPI.

“The estimated glomerular filtration rate (eGFR) was calculated based on serum creatinine levels using the Chronic Kidney Disease Epidemiology Collaboration equation.”

(6) Why not include smoking status in the logistic regression model? And why the analyses for FCV% or FEV1% was included different covariates in the model? How were the covariates of the adjusted model selected?

Thank you for your comments.

We included all covariates including smoking status and modified table 5 and 6.

(7) Did other covariates (such as biochemical indicator) collected in the present study?

Unfortunately, other biochemical indicators were not collected in this study. 

5. Result:

(1) Clinical characteristics of the study population: Paragraph 2, “compared to” may be better than “relative to”.

We changed “relative to” �” compared to” in all sentences.

(2) The detail value of odds ratio and its 95% confidence interval of “1.68-fold” should be presented.

AS you pointed out, we presented 95% CI and p-value.

(3) In the Table 4, why not provide model 1, 2, 3 in the men?

We added model 1 and 2 in the men in table 4.

(4) Univariate or multivariate logistic regression was used in the Table 5 and Table 6?

Table 5 and 6 showed multivariable logistic regression analyses. We modified the titles of the table 5 and 6.

(5) The association between serum uric acid and lowest quartile of FVC% predicted or lowest quartile of FEV1% predicted should be analyzed.

Regarding to FVC% predicted, only in women, serum uric acid level and the prevalence of hyperuricemia were higher in the Q1 group compared to the Q2–4 group.

Regarding to FEV1% predicted, both in men and women, serum uric acid levels were not significantly different between the Q1 and Q2–4 groups. However, the prevalence of hyperuricemia was higher in Q1 than Q2-4 groups, in both men and women.

We described the association between serum uric acid and lowest quartile of FVC% predicted and lowest quartile of FEV1% predicted in result section and presented Table 2 and 3.

(6) How about the stratified analyses of the association between uric acid and the FVC% predicted and FEV1% predicted? 

Unadjusted analysis

men Quintile 1 Quintile 2 Quintile 3 hyperuricemia p

Uric acid (mg/dl) (-5.0) (5.1-5.9) (6.0-7.0) >7 

FVC % predicted 89.6± 0.6 89.2 ± 0.7 90.7 ± 0.6 88.2± 0.8 0.114

FEV1 % predicted 88.9 ± 0.7 88.9 ± 0.7 90.1 ± 0.7 87.3± 0.9 0.143

women Quintile 1 Quintile 2 Quintile 3 hyperuricemia p

Uric acid (mg/dl) (-3.9) (4.0-4.6) (4.7-6.0) >6 

FVC % predicted 93.3 ± 0.5 93.3 ± 0.5 90.9 ± 0.5 86.5 ± 1.3 <0.001

FEV1 % predicted 93.1 ± 0.5 93.8 ± 0.6 92.2 ± 0.5 90.1 ± 1.6 0.039

In men, there is no association between serum uric acid level and FVC% predicted or FEV1% predicted. However, in women, serum uric acid was inversely associated with FVC% predicted and FEV1% predicted.

Theses finding were similar to the results from the comparison of serum uric acid levels between Q1 and Q2-4 FVC% predicted or FEV1% predicted. To avoid redundancy and confusion, we did not add this content additionally. 

6. Discussion:

(1) Paragraph 3: “mean age is” might better than “mean age =”.

 We changed to “mean age is”

 (2) Paragraph 7: The description, “the first study” is inappropriate. 

We changed “first study” to “large scale study”

7. Overall, the quality of date analysis, and the paper writing should be further improved.

As your suggestion, we modified the data analyses and revised the manuscript on the whole in light of the reviewer’s comments.

Reviewer #3: The authors proposed the association between hyperuricemia and low pulmonary function in women not in men. The idea is interesting. However, many detailed information was lacking, for example, about 40% people included in the final analysis, making it difficult to judge the validity of the study findings. My comments are:

1. In the current study, authors defined smoking category as non-smoker, ex-smoker and current smoker. Have authors considered to categorize as active smoker (defined as participants who reported smoking ≥100 cigarettes in their lifetime and who currently smoked) and Passive smokers (defined as those who did not smoke but had been exposed to tobacco smoke either in the workplace or at home during the last week)? If it is possible, it is strongly recommended to change current definition to refine their analysis. Authors also need to explain their definition of “non-smoker” and “ex-smoker” in current study. The “non-smokers” can be defined as participants who smoked <100 cigarettes during their lifetime, had not smoked currently, and had not been exposed to second hand tobacco smoke. The “ex-smokers” can be defined as participants who smoked >100 cigarettes during their lifetime, had not smoked currently, and had not been exposed to second hand tobacco smoke”. There are many references available that authors can refer it to their finding. Considering that smoking may have a long-term effect on SUA/hyperuricemia, the finding may be biased from low smoking rate in women.

We could not change the definition of smoking status, since we used data from KNHNES, which defined smoking category as follows,

Current smoker: smoking ≥100 cigarettes in their lifetime and who currently smoked

Ex-smoker: smoking ≥100 cigarettes during their lifetime, had not smoked currently

Non-smoker: smoking <100 cigarettes during their lifetime, had not smoked currently

2. Since gender is a strong factor for SUA/hyperuricemia, the authors need to explain their finding only in women. By doing so, the readers could get much clear picture of distribution of potential confounders.

Thank you for your comments. We added discussion about the sex difference in the association between serum uric acid and low pulmonary function, as follows.

“In this study, there was sex-difference in the association between serum uric acid and low pulmonary function as well as age. No association was observed between serum uric acid levels and spirometric pulmonary function in men. In contrast, unadjusted or adjusted FVC% predicted and FEV1% predicted were negatively correlated with serum uric acid levels in women, irrespective of age. Regarding to hyperuricemia, in women, hyperuricemia was associated with both lowest quartile of FVC% predicted and FEV1 predicted. However, in men, hyperuricemia was associated with only the lowest quartile of FEV1% predicted, not FVC% predicted. These results suggested that reverse correlation between serum uric acid and pulmonary function was more pronounced in women. However, the mechanism of the age or sex-specific difference between them is still unclear. Previously, although several studies showed that sex differences in the association between hyperuricemia and cardiometabolic risk, there has been no studies about sex differences in the relation between hyperuricemia and lung function [26-29].

One possibility is that there is an inherent difference in uric acid metabolism between the sexes, including a higher fractional urate excretion and lower serum urate concentration in women compared to men [26, 30]. Women may have greater xanthium oxidase activity and reactive oxygen species (ROS) production, resulting in severe vascular inflammation, arterial stiffness, and elevated blood pressure as compared with men. We speculated that estrogen change might play a role in mediation of the association between serum uric acid and lung function. Another possibility is that other cofactors influencing pulmonary function not considered in this study may also be affected by gender.”

 3.By adjusting eGFR in model 2, the association became stronger. Authors need to explain this phenomenon with plausible mechanism in discussion.

Spearman’s correlation was used in order to determine the association between serum uric acid level and FVC% predicted, FEV1% predicted or FEV1/FVC. Age was adjusted in Model 1. In Model 2, age, education, household income (KRW), smoking (non-, ex-, and current smoker), alcohol consumption (none, 1-3/week, and � 4/week), aerobic exercise, BMI (kg/m2), systolic BP (mmHg), fasting plasma glucose (mg/dL), and eGFR (mL/min/1.73 m2) were adjusted. 

Spearman correlation coefficient were likely to be similar in model 1 and 2.

4. Both alcohol consumption and BMI were strong risk factors for SUA/hyperuricemia. In the current analysis, residual confounding by these two factors may be one potential explanation for the current findings. Authors also need to address these issues with further analysis.

Alcohol intake none 1-3/week � ≥ 4/week p

BMI men 24.5 ± 0.2 24.6 ± 0.1 24.7± 0.2 0.787

 Women 24.4 ± 0.1 24.1 ± 0.1 24.4± 0.6 0.204

Unfortunately, we could not find significant association between alcohol intake and BMI in this study. So we could not explain the current finding with residual confounding by these two factors

5. Authors need to address the method of multiple imputation in current analysis and how many percentages of covariates were imputed. If not, authors need to address how to deal with missing values.

Since we excluded subjects who did not undergo uric acid levels and pulmonary function tests from KNHNES which did not include missing values in other demographic and clinical factors as covariates, we didn’t have to deal with missing values.

---

## [Decision Letter · Decision Letter 1]

29 Sep 2020

PONE-D-19-34352R1

Association between Serum Uric Acid and Spirometric Pulmonary Function in Korean Adults: The 2016 Korea National Health and Nutrition Examination Survey

PLOS ONE

Dear Dr. Kim,

Thank you for submitting your manuscript to PLOS ONE. After careful consideration, we feel that it has merit but does not fully meet PLOS ONE’s publication criteria as it currently stands. Therefore, we invite you to submit a revised version of the manuscript that addresses the points raised during the review process. We will accept your paper after clarifying reviewer 3's comment at your discussion.

We look forward to receiving your revised manuscript.

Kind regards,

Sung Kweon Cho

Academic Editor

PLOS ONE

Reviewers' comments:

Reviewer's Responses to Questions

**Comments to the Author**

1. If the authors have adequately addressed your comments raised in a previous round of review and you feel that this manuscript is now acceptable for publication, you may indicate that here to bypass the “Comments to the Author” section, enter your conflict of interest statement in the “Confidential to Editor” section, and submit your "Accept" recommendation.

Reviewer #1: All comments have been addressed

Reviewer #3: All comments have been addressed

2. Is the manuscript technically sound, and do the data support the conclusions?

Reviewer #1: Yes

Reviewer #3: Yes

3. Has the statistical analysis been performed appropriately and rigorously? 

Reviewer #1: Yes

Reviewer #3: Yes

4. Have the authors made all data underlying the findings in their manuscript fully available?

Reviewer #1: Yes

Reviewer #3: Yes

5. Is the manuscript presented in an intelligible fashion and written in standard English?

Reviewer #1: Yes

Reviewer #3: Yes

6. Review Comments to the Author

Reviewer #1: The authors have addressed all my comments well and I have no further comments regarding this revised manusciprt.

Reviewer #3: The sentence of below can be misled readers without references. I strongly recommend to update references. If not, it is needed to be deleted.

"Women may have greater xanthium oxidase activity and reactive

oxygen species (ROS) production, resulting in severe vascular inflammation, arterial stiffness,

and elevated blood pressure as compared with men."

7. PLOS authors have the option to publish the peer review history of their article (what does this mean?). If published, this will include your full peer review and any attached files.

Reviewer #1: **Yes: **Gen-Min Lin

Reviewer #3: **Yes: **Sung Kweon Cho, MD/Ph.D, NCI/NIH

---

## [Author Response · Author response to Decision Letter 1]

4 Oct 2020

Reply to Reviewers

Reviewer #3: The sentence of below can be misled readers without references. I strongly recommend to update references. If not, it is needed to be deleted.

"Women may have greater xanthium oxidase activity and reactive

oxygen species (ROS) production, resulting in severe vascular inflammation, arterial stiffness,

and elevated blood pressure as compared with men."

 Thank you for your comments. As your suggestion, we deleted those sentences to avoid unnecessary misunderstanding.

---

## [Editor Report · Decision Letter 2]

7 Oct 2020

Association between Serum Uric Acid and Spirometric Pulmonary Function in Korean Adults: The 2016 Korea National Health and Nutrition Examination Survey

PONE-D-19-34352R2

Dear Dr. Kim,

We’re pleased to inform you that your manuscript has been judged scientifically suitable for publication and will be formally accepted for publication once it meets all outstanding technical requirements.

Kind regards,

Sung Kweon Cho

Guest Editor

PLOS ONE
---

## [Editor Report · Acceptance letter]

12 Oct 2020

PONE-D-19-34352R2 

Association between Serum Uric Acid and Spirometric Pulmonary Function in Korean Adults: The 2016 Korea National Health and Nutrition Examination Survey 

Dear Dr. Kim:

I'm pleased to inform you that your manuscript has been deemed suitable for publication in PLOS ONE. Congratulations! Your manuscript is now with our production department. 

Kind regards, 

on behalf of

Dr. Sung Kweon Cho 

Guest Editor

PLOS ONE